# NEWTS1.0: Numerical model of coastal Erosion by Waves and Transgressive Scarps

Rose V. Palermo[1,2], J. Taylor Perron[3], Jason M. Soderblom[3], Samuel P. D. Birch[4], Alexander G. Hayes[5], Andrew D. Ashton[6]

[1] U. S. Geological Survey, St. Petersburg Coastal and Marine Science Center, St. Petersburg, Florida 33701, USA
[2] MIT-WHOI Joint Program in Oceanography/Applied Ocean Science & Engineering, Cambridge and Woods Hole, MA, USA
[3] Department of Earth, Atmospheric and Planetary Sciences, Massachusetts Institute of Technology, Cambridge, MA, USA
[4] Department of Earth, Environmental, and Planetary Sciences, Brown University, Providence, RI, USA
[5] Department of Earth, Atmospheric and Planetary Sciences, Cornell University, Cambridge, MA, USA
[6] Department of Geology and Geophysics, Woods Hole Oceanographic Institution, Woods Hole, MA, USA

*Correspondence to*: Rose V. Palermo (rpalermo@usgs.gov)

Abstract: Models of rocky coast erosion help us understand the physical phenomena that control coastal morphology and evolution, infer the processes shaping coasts in remote environments, and evaluate risk from natural hazards and future climate change. Existing models, however, are highly complex, computationally expensive, and depend on many input parameters; this limits our ability to explore planform erosion of rocky coasts over long timescales (100s to 100,000s years) and a range of conditions. In this paper, we present a simplified cellular model of coastline evolution in closed basins through uniform erosion and wave-driven erosion. Uniform erosion is modeled as a constant rate of retreat. Wave erosion is modeled as a function of fetch, the distance over which the wind blows to generate waves, and the angle between the incident wave and the shoreline. This reduced complexity model can be used to evaluate how a detachment-limited coastal landscape reflects climate, sea level history, material properties, and the relative influence of different erosional processes.

## 1 Introduction

Rocky coastlines are erosional coastal landforms resulting from the landward transgression of a shoreline through bedrock. They make up approximately 80% of global coasts (Emery and Kuhn, 1980) and often erode slowly through the impact of waves (Adams et al., 2002, 2005), abrasion by sediment (Sunamura, 1976; Robinson, 1977; Walkden & Hall, 2005; Bramante et al., 2020), and chemical weathering (Sunamura, 1992; Trenhaile, 2001). Rocky coastlines protect coastal communities from erosion and flooding, provide sediment for estuaries, marshes, and beaches, serve as important habitats (such as kelp forests), and support tourism economies. The imprint that each erosional mechanism leaves on the shoreline may be further complicated by sea-level changes, accumulation and redistribution of sediment, heterogeneities in the bedrock, or climate forcings. Wave-driven erosion occurs at a rate proportional to the wave power (Huppert et al., 2020). Therefore, over long time scales, waves tend to erode more exposed parts of coastlines preferentially, blunting headlands while preserving the shapes of sheltered embayments. South Uist, Scotland exemplifies this phenomenon, where the west side of the island is open to the Atlantic Ocean and therefore smoother than the east side, which is relatively protected (Fig. 1c). Uniform erosional processes, like dissolution or mass backwasting, erode at a nearly uniform rate everywhere along a coastline and result in smooth, rounded coastal features punctuated by skewed, pointy promontories or headlands (Howard, 1995). Instances of

dissolution and backwasting include karst lakes found in Florida, USA (Fig. 1a) as well as scarp
retreat due to weathering and backwasting, such as Caineville Mesa, Utah, USA (Fig. 1b).

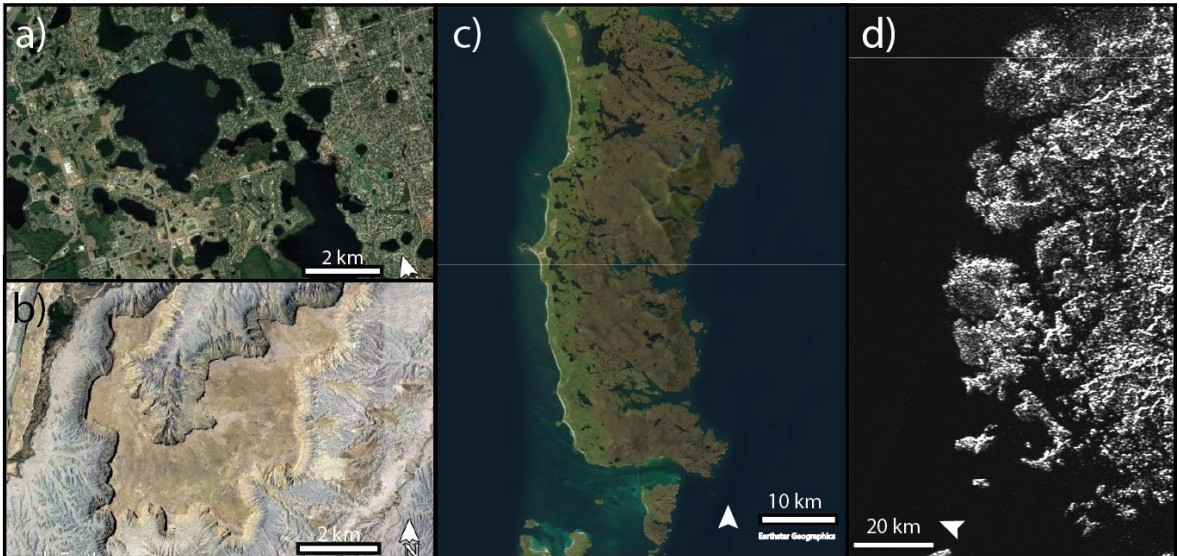


Figure 1: a) Karst lakes in Florida, USA (Map Data: © Google Earth,
Landsat/Copernicus). Lake Butler and the surrounding region. b) Caineville Mesa, Utah, USA
(Map Data: © Google Earth, Landsat/Copernicus). c) South Uist, Scotland (Map Data: Esri
World Imagery, Earthstar Graphics). d) Cassini synthetic aperture radar (SAR) image of Kraken
Mare, Titan (NASA).
Although the relative influence of uniform erosion processes, such as dissolution, and
wave-driven erosion are still being quantified (Trenhaile, 2015), the shape of coastlines may
offer a means to infer dominant processes in remote environments where in situ measurements
are impractical, such as arctic coasts, where local field data are sparse, or remote planetary
bodies, such as Titan (Fig. 1d). A reduced complexity model of long-term, planform evolution of
erosion-dominated coasts can provide insights about the importance of wave erosion relative to
uniform erosion, such as backwasting of permafrost (Günther et al., 2013). Here, we present a
reduced-complexity model of detachment-limited coastal erosion in closed basins, such as lakes
or inland seas, by uniform erosion and wave erosion. We test the model by comparing our
numerical solution of erosion with an analytical solution and test for model result sensitivity to
grid resolution and input parameters. Finally, we describe how this model may be applied
beyond closed basins to open coasts and islands (See Section 5).
2 Background
2.1 Previous Models of Coastal Erosion
2.1.1 Models of wave-driven erosion
Models of rocky-coastline geomorphology have historically focused on the erosion of the
cross-shore profile through sea-level rise (Walkden and Hall, 2005; Young et al., 2014), wave
impacts (Adams et al., 2002, 2005; Huppert et al., 2020), and the competing effects of sediment
abrasion and sediment cover (Kline et al., 2014; Young et al., 2014; Sunamura 2018; Trenhaile,
2019). But recent work has explored the alongshore variability (Walkden and Hall, 2005) and
planform evolution of these features (Limber & Murray, 2011; Limber et al., 2014; Sunamura,
2015; Palermo et al., 2021), with particular focus on either the relationship between planform
morphology and retreat rates following storms (Palermo et al., 2021) or the persistence of an
equilibrium coastline shape consisting of headlands interspersed with pocket beaches due to
variable lithology, grain size, or sediment tools and cover (Trenhaile, 2016; Limber & Murray,
2011; Limber et al., 2014).
Existing models of planform erosion of rocky beaches include 1) a mesoscale (1 to 100
years) alongshore-coupled cross-shore profile model, SCAPE (Walkden and Hall, 2005), in
which waves erode the substrate when the substrate is not armored by sediment and sediment is
transported by waves using linear wave theory; 2) a numerical model of sea-cliff retreat that
focuses on the mechanical abrasion of a notch at the cliff toe and subsequent failure of the cliff
and sediment comminution in the surf zone (Kline et al., 2014); and 3) a numerical model of
headlands and pocket beaches that takes into account wave energy convergence/divergence and
the processes of sediment production and redistribution by waves (Limber at al., 2014).
Previous work on marsh-shoreline erosion considers the heterogeneity of substrate
erodibility using a percolation theory model (Leonardi & Fagherazzi, 2015). In this system, low
wave energy conditions lead to patchy failure of large marsh portions, resulting in a strong
dependence on the spatial distribution of substrate resistance. In contrast, high-wave-energy
conditions cause the shoreline to erode uniformly, such that the spatial heterogeneity in marsh
erodibility does not influence the erosion rate (Leonardi & Fagherazzi, 2015). This ignores
variations in fetch, which can be important for rocky coastal systems.
These previous process-based models are all computationally expensive and require
specific knowledge of sediment and wave characteristics to accurately apply at local scales. To
model systems for which minimal field data are available, or to explore the general behavior of
planform erosion in rocky coasts under a broad range of conditions, a reduced-complexity model
(Ranasinghe, 2020) is necessary.
2.1.2 Models of uniform erosion
Howard (1995) modeled the retreat of a closed basin scarp as a uniform erosion process.
Howard's approach identifies gridded domain points as either interior or exterior to the
escarpment and erodes the escarpment edge at a constant rate in all directions originating from
adjacent points (Howard, 1995). In his model experiments, the escarpment retreats uniformly
toward the interior of the domain from the exterior. This uniform scarp retreat is analogous to
coastline retreat in response to dissolution of a uniform substrate. Although Howard's model was
designed for a different, subaerial system, uniform erosion of a closed-basin liquid shoreline can
be described with the same process law, as we assume the planform shoreline also erodes at the
same rate in all directions.
Shorelines formed by dissolution in karst landscapes have received some attention,
mostly in the context of cave collapse features or sinkholes (Johnson, 1997; Martinez et al.,
1998, Yechieli et al., 2006). However, most research has focused on the initial formation of these
features; studies of the long-term retreat of coastlines due to dissolution are focused on the
meter-scale erosion of coastal notches through mechanical and biochemical erosion and by
dissolution (Trenhaile 2013; Trenhaile, 2015) and to our knowledge have not been evaluated
over a larger spatial scale.

## 3 Model

We developed the Numerical model of coastal Erosion by Waves and Transgressive
Scarps, V1.0 (NEWTS1.0) (Palermo et al., 2023) to study the planform-shoreline erosion of
detachment-limited coasts by waves, uniform erosion, or a combination of these processes. This
reduced-complexity model can be used to explore long-term (thousands to millions of years)
trends in landscape evolution that result from these processes across the appropriate sea- or lake-
level change conditions. Uniform erosion includes dissolution or mass backwasting and is
modeled with a spatially uniform rate of shoreline retreat, which generally smooths the coastline
and generates cuspate points where promontories are eroded. Wave erosion occurs in proportion
to the wave energy that the coastline is exposed to and to the angle of incidence of the incoming
waves, such that the erosion rate depends on the wave energy in the cross-shore direction per
unit of length along the coast (Komar, 1997; Ashton & Murray, 2009; Huppert et al., 2020).
Coastlines that have larger exposure (larger fetch) experience higher wave energy and therefore
faster wave erosion. We model this energy-dependent erosion by computing the fetch of every
incident wave angle that may impact a given point on the shoreline and weighting this fetch by
the cosine of the angle between the incident wave crests and the shoreline. Mathematically, this
is equivalent to the dot product of the direction of wave travel and the direction normal to the
shoreline.

## 3.1 General description and model setup

## 3.1.1 Model domain and structure

3.1.1.1 Model domain
The domain of the model (Fig. 2) is a grid discretized into $N_x$ cells in the $x$ direction and $N_y$
cells in the $y$ direction, with cell spacings $\Delta x$ and $\Delta y$, such that $x_i = i\Delta x$ and $y_j = j\Delta y$. The
value of each grid cell, $z_{i,j}$, corresponds to the landscape elevation. The boundaries of the grid
are periodic. Each cell in the domain is defined as either liquid or land based on its elevation
relative to sea or lake level. The model could apply to lake level in closed liquid bodies or sea
level in semi-closed seas or open coasts. For simplicity, in this manuscript we will use "lake" to
refer to the liquid bodies, "lake cell" refers to cells occupied by liquid, and "lake level" refers to
the elevation of the liquid level. Cells below lake level are fixed and do not erode. Shoreline
cells, defined as land cells directly adjacent to liquid, may be eroded by coastal processes
through uniform erosion and wave erosion. Lake level is an input to the system that the user can
vary throughout a model run.


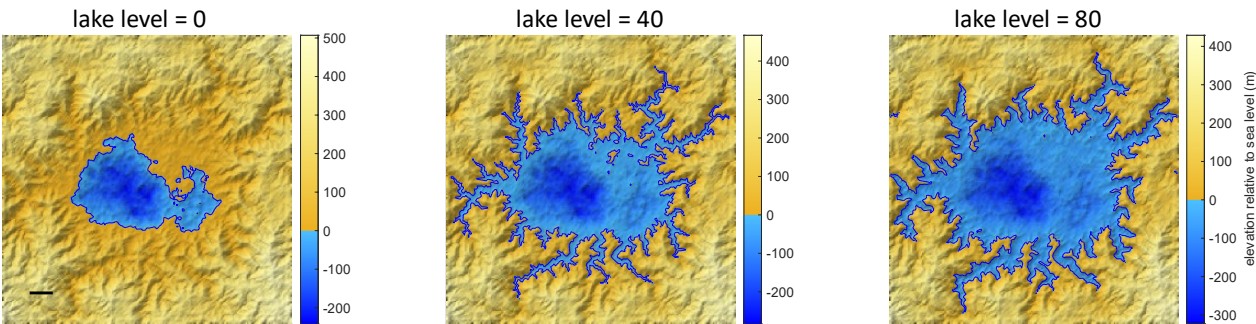

Figure 2: Example model domain with a lake level of a) 0 m, b) 40 m, and c) 80 m. This domain
is used in Figs. 4 and 5.
3.1.1.2 Identification of liquid body and shoreline cells
Boundaries in the grid are identified using pixel connection definitions of either 4-connected,
in which connections occur only across edges, or 8-connected, in which connections occur either
across edges or at corners. Liquid cells that are 8-connected to each other comprise the same
liquid body. The liquid body could represent an sea or lake, so for simplicity we call a liquid cell
a "lake cell" and a liquid body a "lake" in this manuscript. Islands are defined as groups of land
cells that are surrounded by liquid cells. Lakes can also occur inside islands and islands inside
these lakes, so we define a lake hierarchy to identify and model each lake individually. The first
level in this hierarchy is the land that is connected to the border of the domain. First order lakes
are lakes that are immediately surrounded by this land that extends to the border of the domain.
A first order island is immediately surrounded by a first order lake. A second order lake is
surrounded by a first order island, and so on. This continues such that Nth-order islands are
surrounded by Nth-order lakes, and Nth-order lakes are surrounded by N-minus-one-order
islands. This hierarchy allows us to identify and isolate unique lakes, which will be important
when we consider wave-driven erosion.
3.1.1.3 Cellular grid erosion
Each cell starts with an initial strength, $S_{init}$, (see Sections 3.1.3 to 3.3) which is depleted
according to a rate law associated with each coastal process until reaching 0 (see Sections 3.2
and 3.3), at which point the cell erodes. Coastal erosion occurs on shoreline cells, defined as land
cells adjacent to liquid cells, and decreases the elevation of those cells by a specified depth of
erosion, $d_e$, which is user specified. For cells eroded by coastal processes, $z(t) = z(t-1) -$
$d_e$, where $t$ is model time. For uniform erosion, $d_e$ is conceptualized as the scarp dissolution
depth. For wave erosion, is conceptualized as a wave base. Shoreline cells become lake cells
once eroded. To avoid numerical artifacts associated with the time discretization, the timestep
must be set such that the amount of erosion per iteration is a small fraction of the total cell size.
In practice, we set the time step to erode less than 1/10[th] of a cell at a given time given the cell
spacing and rate law. The model run terminates if a lake cell becomes adjacent to a boundary cell
because the wave erosion model requires a closed coastline.

3.1.1.4 Order of operations
During each timestep, erosion occurs according to three steps, if enabled: 1) Sea- or lake-
level Change, 2) Wave Erosion, and 3) Uniform Erosion (Fig. 3). Here we describe the general
model components and simulation procedure. The governing equations for Uniform Erosion and
Wave erosion are outlined in more detail in sections 3.2 and 3.3, respectively.

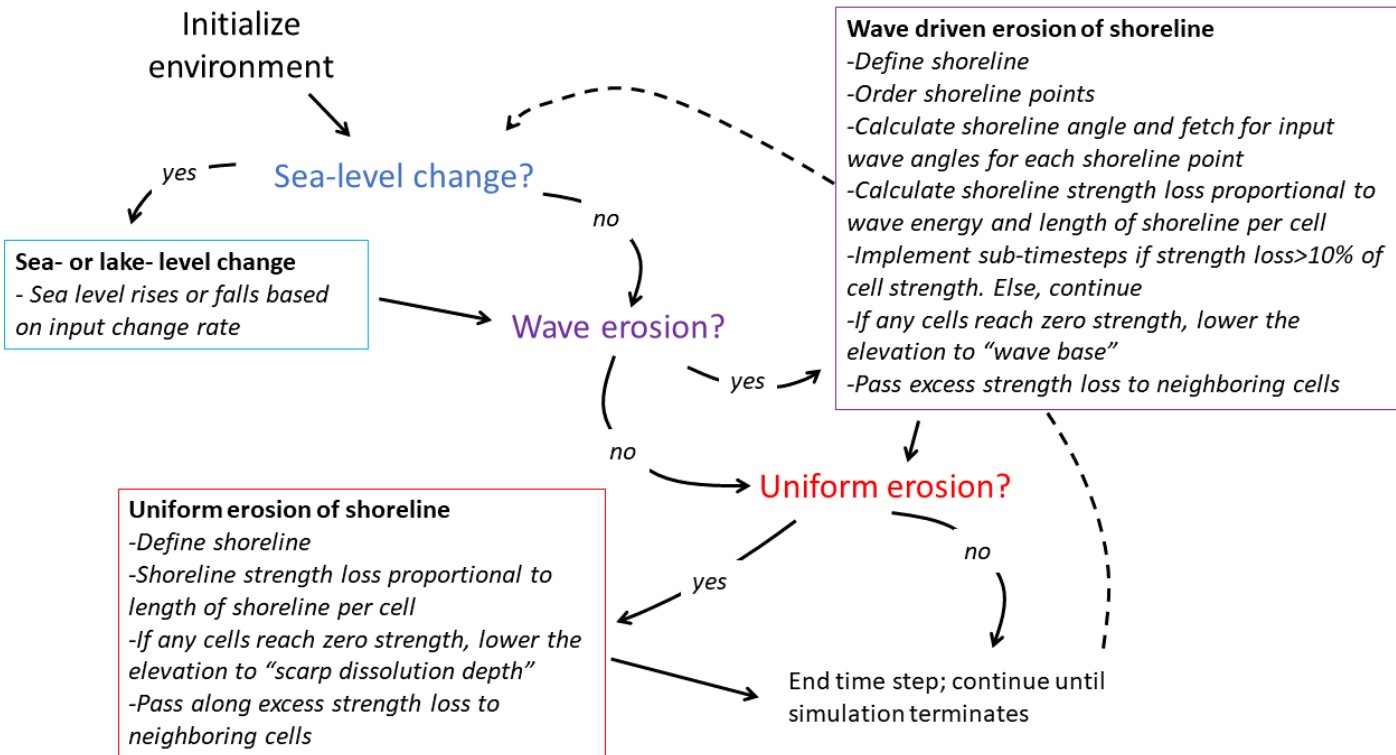

Figure 3: Model structure showing the time loop in which the model 1) updates sea- or lake-level
change, then calculates shoreline erosion due to 2) waves and 3) uniform erosion processes.
The first operation of the model is lake-level change. The lake level changes as an input rate
or according to an input lake- level curve. The new lake level is used to define the lake(s) and
shoreline(s) (Section 3.1.1.2 and 3.1.2).
Next, wave erosion of the shoreline(s) occurs as a function of the fetch—the open-water
distance wind and waves travel before reaching a point on the coast—and the angle between the
wave crests of the incident waves, $\varphi$, and the azimuth of the shoreline, $\theta$ (Section 3.3). In this
module, the shoreline is first identified and traced such that shoreline cells are ordered in a
counterclockwise direction. The shoreline is then used to calculate the shoreline angle, incident
wave angle, and associated fetch at each cell along the shoreline (Section 3.3.1). The elevation of
eroded shoreline cells is lowered, their labels are changed to liquid cells as appropriate, and the
shoreline is updated (See Section 3.4, Fig 5). This approach considers sediment removal as
instantaneous. Future variations of the model could consider the erosion also as a function of the
height of the material being eroded or the excavation rate of weathered rubble.
Finally, uniform erosion of the updated shoreline occurs (Section 3.2). Here, the shoreline
erodes as a function of the alongshore length of the shoreline as measured along cell boundaries
(Section 3.1.2 and 3.2). And again, the elevation of eroded shoreline cells is lowered, the labels
of eroded cells are changed to liquid cells, and the shoreline is updated.
## 3.1.2 Defining the shoreline

There are two options for defining shoreline cells: the 8-connected case, in which successive
land cells along the shoreline may border one another either at cell edges or at cell corners (Fig.
4a), or the 4-connected case, in which successive land cells along the shoreline may border one
another only at cell edges (Fig. 4b). In the case of an 8-connected shoreline, shoreline cells only
border liquid cells at cell edges (Fig. 4a), whereas shoreline cells in a 4-connected shoreline can
border liquid cells at cell edges or at cell corners (Fig. 4b). We choose $\Delta x$ and $\Delta y$ to be small
enough to represent the relevant features of the shoreline. If lake-level change occurs in the
simulation, the relevant features in the landscape should be taken into account when choosing $\Delta x$
and $\Delta y$. Here, we present simulations where $\Delta x$ and $\Delta y$ are equal. The model can operate with
different $\Delta x$ and $\Delta y$; however, there could be resulting differences in error which have not been
tested.

a)

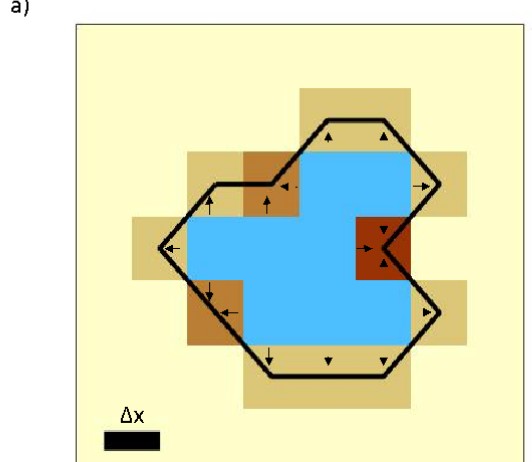

b)

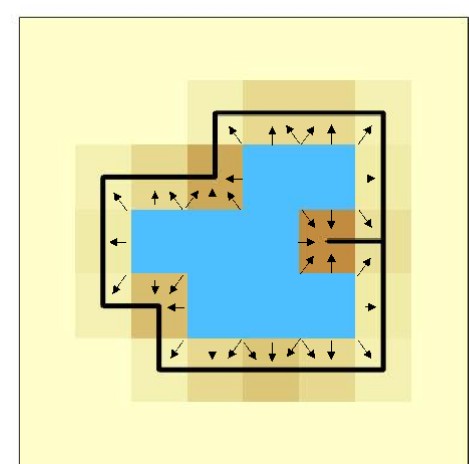


Figure 4: Shoreline cells and associated strength loss weighting for a shoreline that is a) 8-
connected or b) 4-connected. Arrows point in the direction of erosion into each shoreline cell
from neighboring lake cells. Increasing darkness of shoreline cells indicate increasing strength
loss weighting.

The shoreline cells need to be ordered so that the lake can be represented as a polygon for
the fetch computation. To order the shoreline cells in closed loops, we start at the first indexed
shoreline cell of the longest shoreline and move counterclockwise to find the next shoreline cell.
Once a sequence of the first 3 cells is repeated, the loop is closed and the shoreline is deemed
complete. Any remaining shoreline cells that do not lie on this loop represent the shoreline of a
separate first-order lake, or of an island or higher order lake contained within the lake. Next,
ordering the shorelines of the islands contained within the current lake begins on the first
remaining shoreline cell. We repeat this process until all land cells bordering liquid are included
in a closed shoreline. When there are multiple first-order lakes in a landscape domain, the
shorelines for each lake and its enclosed islands are ordered one at a time.

### 3.1.3 Cell strength and coastal erosion processes

All cells start with an initial strength, $S_{init}$, which represents how difficult it is to erode
the land (Equation 1). We model the domain as having uniform strength in both planform space
and elevation, but this could easily be extended to a scenario with heterogeneous strength. The
strength of a cell is initialized as a reference strength, $S_0$, multiplied by the ratio between the cell
area, $A = \Delta x \Delta y$, and a reference cell area, $A_0 = \Delta x_0 \Delta y_0$, with reference spacing $\Delta x_0$ and $\Delta y_0$
(Equation 1). The reference strength and area nondimensionalize strength and maintain
proportions that mitigate discretization bias. The magnitude of these values can be chosen by the
user.

$$S_{init} = S_0 \frac{A}{A_0} \tag{1}$$

Strength is lost from each shoreline cell at a rate that depends on the exposed perimeter of
the cell and an erosion rate law specific to either uniform erosion or wave erosion processes.
Change in strength is grid-independent for grids sufficiently fine to satisfy model stability
because the strength is initialized with a reference cell area in proportion to the parameterized
cell area. To mitigate discretization bias, $\Delta x$, $\Delta y$, and $\Delta t$ must be sufficiently small that $\Delta t$ is less
than the time to completely erode a cell (See Sections 3.2 and 3.3), and that $\Delta x$ and $\Delta y$ properly
represent the shoreline morphology. In practice, we choose $\Delta x$ to be equal to $\Delta y$.
As time progresses, each shoreline cell loses strength until failure, $S_{i,j} = 0$, at which
point the cell has eroded. It is possible for the strength loss in one time step to exceed the
remaining strength of the cell. When this occurs, the excess time spent eroding the cell is passed
along to all new shoreline neighbors of the eroded cell, representing the time of erosion that
neighboring cell will incur after the erosion of the original shoreline. If a new shoreline cell is
inheriting excess time from multiple neighbors, the mean excess time is used to compute the
strength loss. In our simulations, taking the mean of the excess time resulted in the least grid
bias.
Modeled erosion could be underestimated or redistributed improperly if the strength loss
for an eroding cell is consistently large relative to the initial strength of the domain. The
shoreline would then not update with the newly exposed cells, rather constantly passing strength
loss to its neighbors, and inaccurately characterizing the morphology. We implement a sub-
timestep routine to capture the effect of the changing shoreline within a single timestep when the
strength loss of any shoreline cell in the domain exceeds a certain threshold of the initial
strength, $\alpha$, which ranges between 0 and 1. In the modified time-step routine, the damage is
computed and the shoreline updated in sub-timesteps, which segments the time-step and allows
erosion to occur in smaller increments.

## 3.2 Uniform erosion model

The rate of shoreline retreat by uniform erosion is set by an erodibility coefficient, $k_{uniform}$ (Eq. 2). Strength loss due to uniform erosion occurs as a function of the amount of shoreline in contact with the lake for a given cell, represented as the number of 4-connected sides, and 8-connected corners, $c$, in contact with lake cells (Eq. 3; Fig. 4). Because the diagonal of the cell is longer than the side by a factor of $\sqrt{2}$, it would take $\sqrt{2}$ times longer for a shoreline to retreat across a cell diagonal than in the perpendicular direction. To correct for this in our model, the strength loss computed from an exposed corner is $\sqrt{2}/2$ as much as the strength lost from an exposed side.

$$\frac{dx}{dt} = k_{uniform},\tag{2}$$

$$\frac{\Delta S_{i,j}}{S_0} = -k_{uniform}\left(s_c + \frac{\sqrt{2}c}{2}\right)\frac{\Delta x}{\Delta x_0}\Delta t,\tag{3}$$

## 3.3 Wave erosion model

Wave erosion occurs at a rate determined by a wave erodibility coefficient, $k_{wave}$ [m·yr$^{-1}$], and the wave energy in the cross-shore direction, $E$ (Eq. 4). The wave energy depends on the wave height, $H$, and the angle between the wave crests of the incident waves, $\varphi$, and the azimuth of the shoreline, $\theta$ (Eq. 5). Wave height scales with fetch, $F$, such that $H \propto \sqrt{F}$ (Hasslemann, 1973; Smith and Waseda, 2008). Therefore, we use fetch to approximate the wave energy density for a wave from a given direction on a coastline (Eq. 6). The use of wave energy implies the assumption of single-period waves.

$$\frac{dx}{dt} = k_{wave}E \ ,\tag{4}$$

$$E = \frac{1}{16}\rho g H^2 cos\,(\varphi - \theta),\tag{5}$$

$$E \propto \rho g F cos\,(\varphi - \theta),\tag{6}$$

The strength loss of a cell due to waves can be described as

$$\frac{\Delta S_{i,j}}{S_0} = -k_{wave}(s_c + \frac{\sqrt{2}c}{2})\int_{\varphi=0}^{2\pi} F(\varphi)cos\,(\varphi - \theta)d\varphi\frac{\Delta x}{\Delta x_0}\Delta t.\tag{7}$$

If the strength loss in a time step exceeds a parameter-set threshold, a sub-timestep routine is implemented. Because the fetch calculation is the costliest step of the model, in this sub-timestep routine, we estimate the fetch weighting by interpolating the fetch of the nearest neighbor shoreline cells. This avoids additional costly fetch computations during the sub-timestep updates and allows us to approximate erosion driven by waves in a way that limits error without slowing down the model simulation.

## 3.3.1 Modeling wave energy density

The rate of strength loss of each shoreline cell is proportional to the wave energy density.
We model the wave energy density to be proportional to the fetch and the cosine of the angle
between the incident wave crest and the shoreline (Fig. 5). To compute this quantity, we measure
the fetch in all directions around the shoreline, in increments of $d\varphi$, for each shoreline cell. For
each direction, we extend a ray from the cell center in the direction $90° - \varphi$ and step along the
ray in increments of a distance δ until reaching the opposite shore. This modeling approach does
not consider the effects of shoaling or refraction, so waves that would approach from beyond 90°
are not considered. When the ray extends past the opposite shoreline, we take one step back and
define this point as the intersection. The distance between this intersection and the originating
shoreline cell center is the fetch in the direction from which a wave would propagate (Fig 4b).
The length of fetch may be truncated at an input maximum length which would represent the
distance at which waves saturate and do not continue to grow. To calculate the amount of
strength loss each cell incurs, we compute the area of a polygon defined by the ray-shoreline
intersections for that cell (Fig. 5a). We call this area the "fetch area." The length of the ray in
each direction is then weighted by the cosine of the angle between the shoreline and the incident
wave crest, $\varphi - \theta$ (Fig. 5a). The area of the polygon defined by these cosine-weighted fetch
lengths is computed and called the "wave area." The wave area for each point on the shoreline
approximates the integral in Eq. 7.

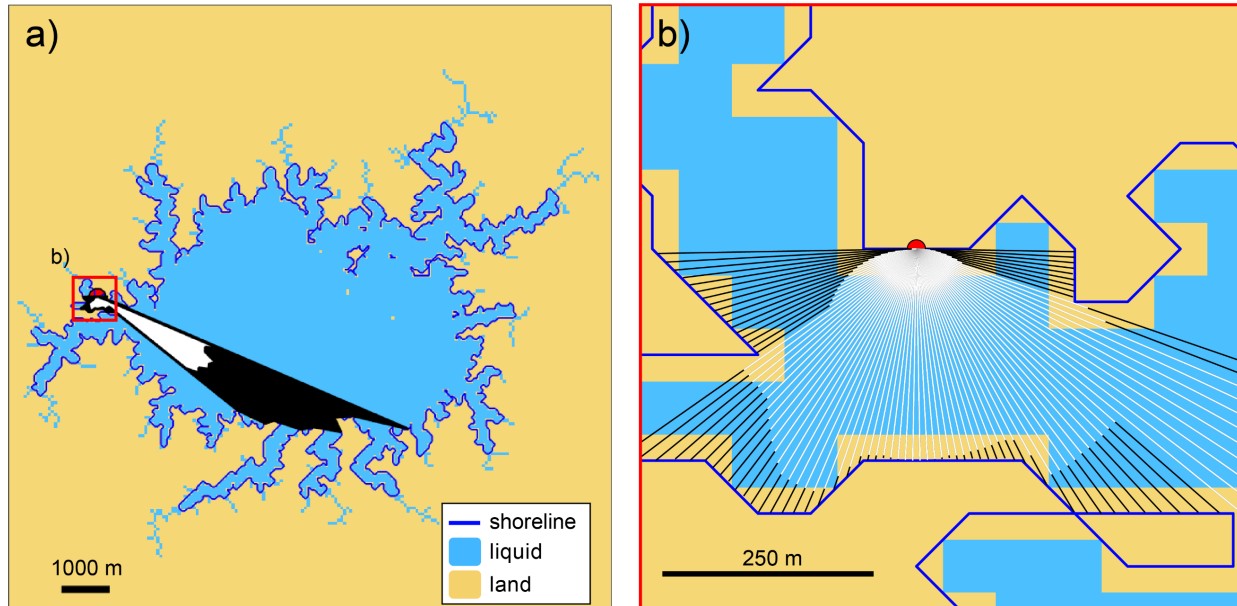


Figure 5: a) Fetch area (black) and wave area (white) computed for a point (red circle) on a
typical model shoreline (blue). The area shown in b) is outlined in red. b) Zoomed-in view of
fetch line-of-sight rays (black) and angle-weighted line-of-sight rays (white) computed for the
same point. In this example, $d\varphi = 2°$ and the ray step size, δ = 0.05 m.
3.4 Model output
The model can be initialized with any user defined topographic model. In the simulations
presented here, we initialize the grid with a synthetic topography consisting of a pseudo-fractal
surface with variance of 10,000 superimposed on an elliptical depression with a depth of 25% of
the domain relief and eroded by river incision to 95% of the initial terrain relief using a
landscape evolution model (Perron et al., 2008, 2009, 2012). We then flood the domain by
raising lake level by 40 m. The model of shoreline retreat by uniform and wave erosion is then
applied to the domain. Here, we show examples of an initial landscape eroded by either wave
erosion or uniform erosion, to illustrate separately the effects of the two erosional mechanisms in
the model (Fig. 6). However, all model components may be run in combination. We do not
provide examples of combined uniform and wave erosion models here.
The initial shoreline exhibits a dendritic shape due to flooding of the incised river valleys
(Fig. 6). Through time, the uniform erosion model drives shoreline retreat at the same rate
everywhere around the perimeter of the lake, resulting in widening valleys and increasing the
pointedness of promontories or headlands (Fig. 6). The overall shape of the lake is maintained,
but becomes smoother and tends toward circular. In the case of wave erosion, the river valleys
erode slowly while the exposed parts of the coast erode more rapidly (Fig. 6). The embayed river
valleys largely maintain their shapes, whereas the central, high-fetch portion of the coast grows
larger and smoother.

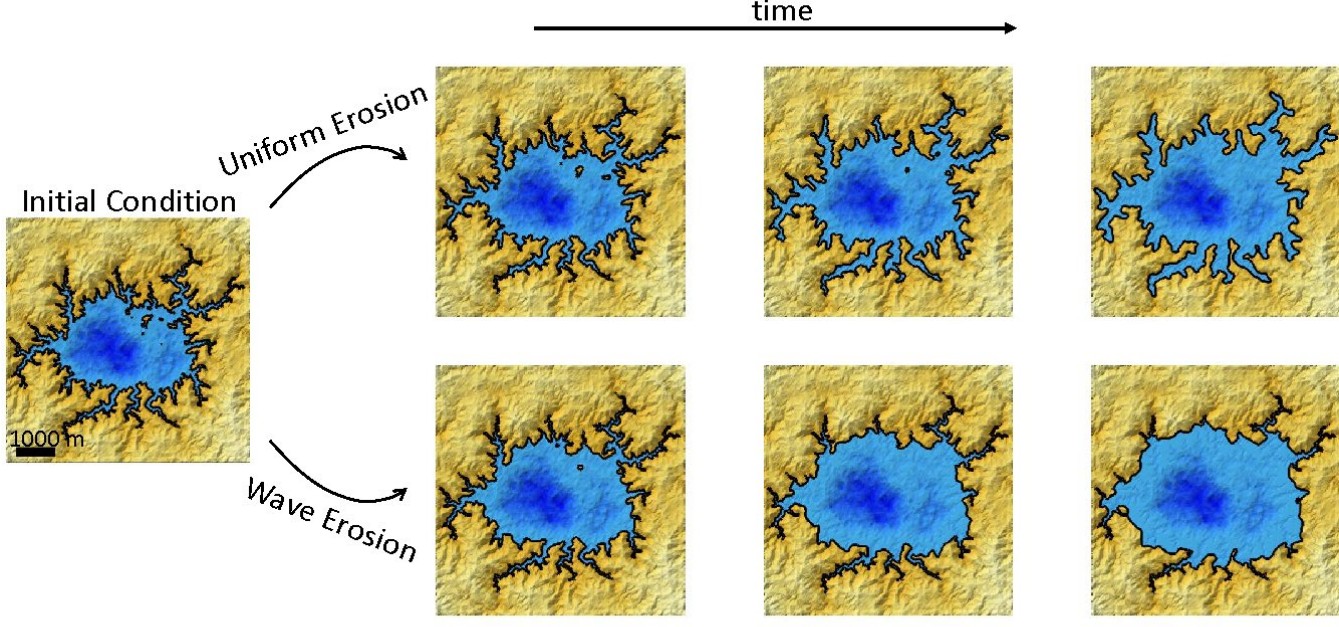

Figure 6: Shaded relief maps of example model simulations of uniform erosion and wave erosion
through time, starting from the same initial condition. Blue color indicates liquid cells, with
darker blues indicating deeper depths. Gold color indicates land cells, with lighter shades
indicating higher elevations. Black lines trace shorelines. Erodibility coefficients are $k_{wave} =$
$k_{uniform} = 0.00001$ m·yr$^{-1}$. Uniform erosion (top) results in greater overall smoothness that is
punctuated by pointy headlands, whereas wave erosion (bottom) results in blunted headlands,
smooth open sections of coast, and preservation of sharp features in sheltered areas. Landscape
time-steps shown correspond to similar amounts of erosion between wave and uniform
examples. The shoreline is defined as 4-connected in these examples.

To test our model performance, we compare the planform morphologies of model output
with example shorelines that have known geomorphic processes. While long term coastal cliff
retreat rates could be determined using dating techniques at local field sites (Hurst et al., 2016;
Bossis et al., 2024), more detailed testing of the model would require recreation of plan-view
shape at a broader scale. Because long-term changes in planform morphology during retreat of
bedrock coastlines are generally too slow to be measurable with historical aerial and satellite
images, the data needed to fully validate this model are not presently available. Nonetheless, a
visual comparison can be drawn between coastal features found on Earth and the coastline
shapes generated by each end-member erosional mechanism in the model, which is the main goal
of our modeling approach. These shorelines exhibit the same overall smoothness, punctuated by
sharp headlands, as is seen in the shorelines formed by uniform erosion in our model (Fig. 6).
Although it is beyond the scope of this paper, output from this model could be used to
quantitatively describe shoreline morphologic differences driven by wave and uniform erosional
processes or signatures of sea- or lake- level changes.
A bedrock lake that has been eroded recently by waves is exemplified by Lake Rotoehu,
New Zealand (Fig. 7c). In these examples, we observe blunted headlands and smooth, rounded
stretches in open sections of coast, and crenulated shorelines in more protected areas of coast –
similar to the shorelines formed by wave erosion in our model (Fig. 6).
Figure 7: a) Lake Rotoehu, New Zealand (Map Data: © Google Earth, CNS/Airbus). b) Plitvice
Lakes, Croatia (Map Data: © Google Earth, DigitalGlobe).
## 4 Model tests
## 4.1 Comparison with analytical solution and sensitivity to shoreline connectedness

For the simple case of an initially circular shoreline, we compute the shoreline evolution
analytically and compare this known solution with our numerical model results. For the uniform
erosion case, the rate at which the radius of a circle increases, $\dot{r}$, is equal to the constant of
erosion, in this case $k_{uniform}$.
$$\dot{r}(t) = k_{uniform} \qquad (8)$$

Therefore, the radius, $r$, at time, $t$, and initial radius, $r_0$, for uniform erosion is:
$$r(t) = r_0 + k_{uniform}t \qquad (9)$$

For wave erosion, the rate of increase of the radius, $\dot{r}$, depends on the constant of erosion, $k_{wave}$,
and the integral of the fetch, $F$, at each angle between the incoming wave crest and the shoreline,
$(\varphi - \theta)$ in all directions around the circle:
$$F(\varphi) = r\sqrt{2(1 + \cos{(2(\varphi - \theta))}} \qquad (10)$$

$$\dot{r}(t) = \frac{k_{wave}}{2} \int_{-\frac{\pi}{2}}^{\frac{\pi}{2}} (F(\varphi)\cos(\varphi - \theta)\,)^2 d\varphi \qquad (11)$$

Computing this integral simplifies to:

$$\dot{r}(t) = k_{wave}\frac{3\pi}{4}r(t)^2 \tag{12}$$

Therefore, the radius, $r$, at time, $t$, for wave erosion is:

$$r(t) = \frac{r_0}{1 - r_0 k_{wave}\frac{3\pi}{4}t} \tag{13}$$

We use the analytical solution for the radius through time for each case to calculate the
shoreline position and area of the circular lake as it is eroded by either uniform or wave erosion.
To compute the relative error of the numerical model, a test circular lake is eroded for 17,400
years, resulting in approximately 20% and 25% increase in lake area for wave and uniform
erosion, respectively, and compare this to the analytical solution.
Because the model operates on a rectangular grid, some amount of distortion of a circle is
expected. While this distortion cannot be avoided entirely by increasing the grid resolution,
increasing it can reduce the error in the shoreline shape by allowing the shoreline to retreat in
finer increments. A fine grid, however, comes at increased computational cost. The spatial
resolution, $\Delta x$ and $\Delta y$, should be chosen to be small enough to represent the features of the
shoreline, but large enough to keep computational costs reasonable.
We perform these simulations for uniform and wave erosion with both 4-connected and
8-connected versions of the model (Fig. 4). The 4-connected model performs significantly better
than the 8-connected model, as shown by the relative error in lake area. The 4-connected case
maintains relative error less than 2% throughout the simulation whereas the error in the 8-
connected model increases roughly linearly with time, ending at approximately 7% (Fig. 7a). The
distortion is worse in the 8-connected case for both uniform erosion and wave erosion, and
systematically worse in the diagonal directions (Fig. 7b,c). This analysis suggests that grid bias is
a more important source of error in the model than spatial discretization.

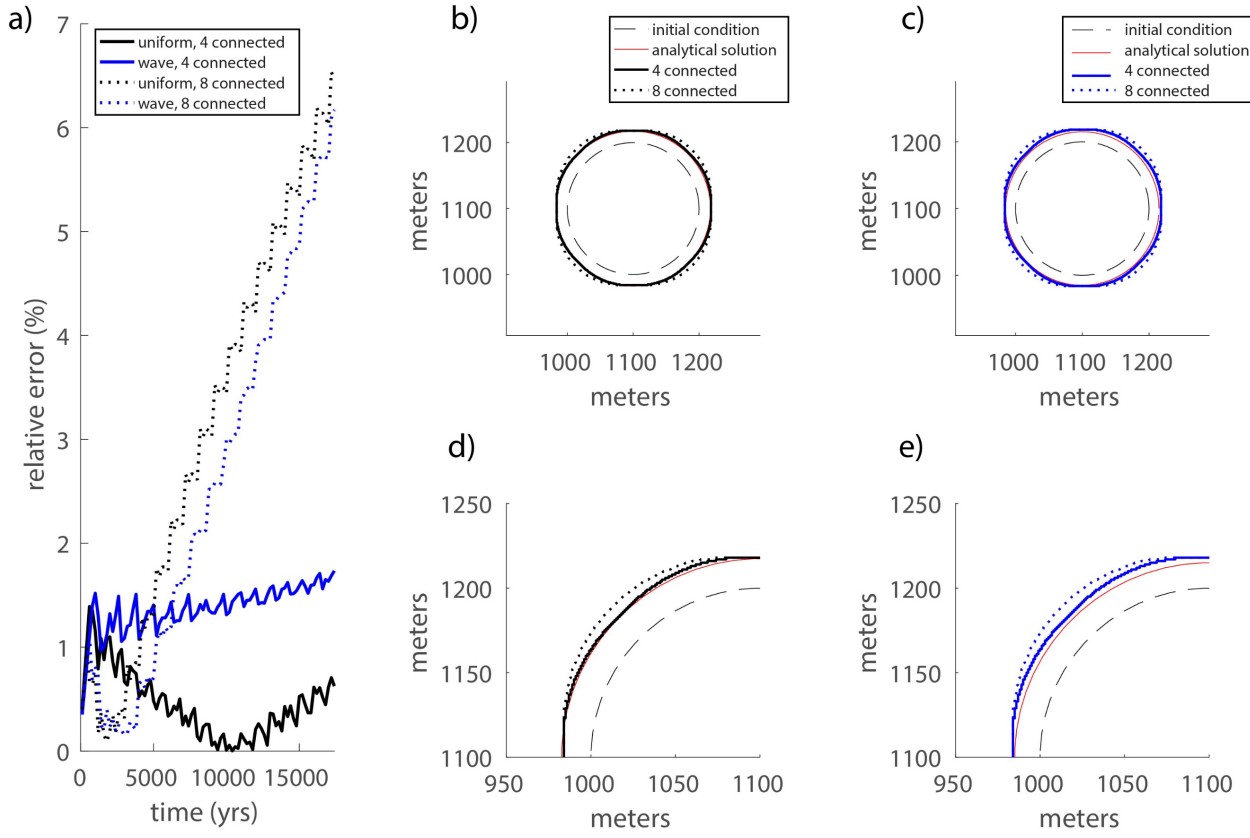


Figure 7: a) The error in lake area through time of an initially circular lake relative to the
analytical solution for 4-connected (solid) and 8-connected (dotted) models of uniform erosion
(black) and wave erosion (blue). The initial condition (dashed), analytical solution (red), and
modeled 4-connected and 8-connected shorelines at time=17400 are shown for b) uniform
erosion and c) wave erosion, with zoomed in results shown for d) uniform erosion and e) wave
erosion.
## 4.2 Resolution sensitivity
### 4.2.1 Grid resolution
Although the grid resolution affects the size of the features that can be resolved in the
landscape, it does not substantially affect the amount of coastal erosion. As discussed above, the
strength loss in this model is insensitive to grid resolution, $\Delta x$, and time step, $\Delta t$, assuming that
$\Delta x$ is fine enough to resolve the features of interest and that $\Delta t$ is small enough to limit erosion
to less than the maximum cell strength in a single time step. The total amount of strength in the
domain is independent of $\Delta x$ because the number of cells is proportional to $\Delta x^{-2}$ and the
strength of each cell is proportional to $\Delta x^2$. The damage in each time step is independent of $\Delta x$
because the number of cells on the shoreline is proportional to $\Delta x^{-1}$ and the damage per cell is
proportional to $\Delta x$.
### 4.2.2 Threshold strength parameter
The threshold strength parameter, $\alpha$, was introduced to prevent excess strength reduction
from being neglected when a cell has less strength than is depleted in a timestep. A smaller
threshold strength parameter results in a more frequent application of the sub-timestep routine
and smaller sub-timesteps. With a less stringent threshold strength parameter (>0.05), the
shoreline may erode more than the analytical solution in a time step, leading to a positive slope
in the relative error in strength against the threshold strength parameter (Fig. 8).

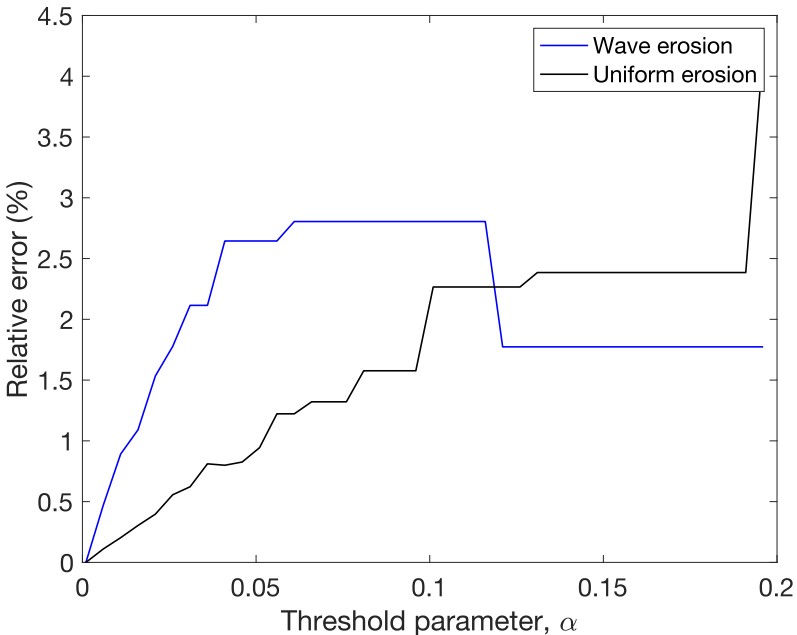


Figure 8: Error in total strength reduction as a function of the threshold strength parameter,
expressed as a percentage of the error for the smallest value of the threshold strength parameter,
for the initial condition in Fig. 6 eroded over one time step by uniform erosion (black) and wave
erosion (blue).

## 4.3 Fetch ray angular and distance increments

We test the sensitivity of the fetch-area calculation to the angle between rays, $d\varphi$, and the
ray step size, $\delta$. This test allows us to analyze the error in fetch of a typical model due to these
parameters. The error measurements provide a basis for selecting an angle between rays and a
ray step size that optimize the trade between computational time and model accuracy.
We compute the error in fetch area over a range of ray angles and step sizes. With a fixed
ray step size of $0.05\Delta x$ (the nominal step sized used in our simulations), we compute the fetch
error for each shoreline cell over a range of $0.012°$ to $10°$, corresponding to 30,000 and 36 rays,
respectively. With a fixed ray angle of $2°$ (the nominal ray angle used in our simulations), we
compute the relative fetch error over a range of ray step sizes between $0.01\Delta x$ to $\Delta x$. The fetch-
area error of each cell is computed relative to the fetch area of the finest resolution in each
parameter: $2°$ between rays and a ray step size of $0.05\Delta x$ (Fig. 9). The error, as well as the
standard deviation in errors, in each scenario converges to zero, indicating that as the angle
between rays and the ray step size become small the fetch area converges to a constant value.

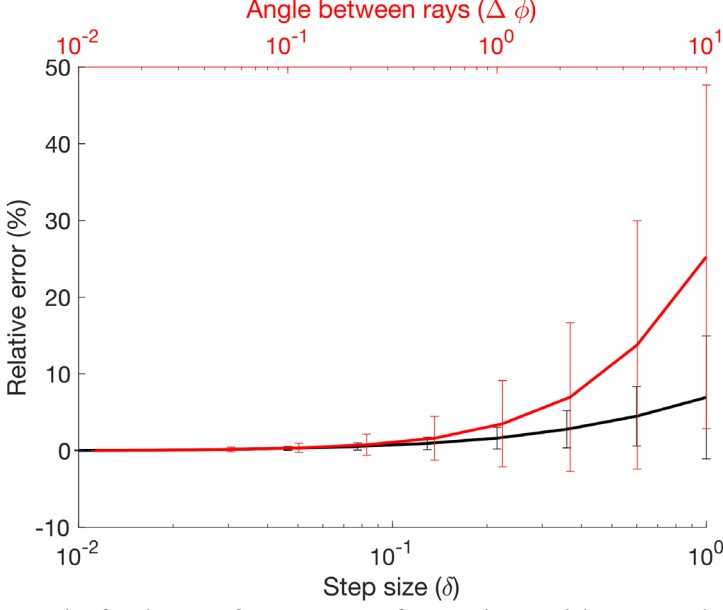

Figure 9: Relative error in fetch area for a range of step sizes with ray angle of 2° (black) and for
a range of ray angles with step size of $0.05\Delta x$ (red).
## 5 Discussion and Conclusions
In this paper, we present NEWTS1.0, a cellular model of coastline erosion in detachment-
limited environments by uniform erosion and by wave erosion. For uniform erosion, the
coastline erodes at a constant rate everywhere along the shoreline. For wave-driven erosion, the
coastline erodes as a function of the fetch and the angle between the incident waves and the
shoreline.
While our uniform erosion rate law is similar to that of Howard (1995), our modeling
approach is different. Because there are multiple mechanisms that may erode a coast in our
model, memory of the strength loss of the substrate is necessary. Rather than rays extending at a
constant rate from the interior points representing retreat as is done in Howard's 1995 model, the
strength of shoreline (or scarp edge) points is reduced by an amount proportional to the number
and direction of neighboring lake cells.
Our wave erosion model contains a dependence on wave energy like in other models
(Walkden and Hall, 2005; Limber et al., 2014), but simplifies the influence of sediment and other
factors to a constant. This simplification is useful for locations without readily available grain
size or sediment cover data, and to investigate the long-term influence of these processes.
However, a limitation of this simplified approach is the implicit assumption of a single wave
period when using wave energy rather than wave power in the wave erosion rate law (Equations
4-6). Future work could extend the capabilities to include consideration of wave period.
Our model is also unusual among coastal erosion models in that it evaluates multiple
closed coastlines (or lakes) in a landscape domain rather than a single reach of open coastline,
and that it focuses on the planform morphology of eroding rocky closed-basin shorelines. A
limitation of this model is that sediment redistribution is not included in the erosion rate laws and
there is no sedimentation along the coast. Sediment abrasion and cover could be incorporated in
future versions of our model through a spatially heterogeneous and time-dependent erodibility
coefficient, $k$; however, this would likely require parameterization from field data.
While this model is currently configured to simulate the erosion of closed basins, such as
lakes or inland seas, modifications could be made to evaluate open stretches of coast. The two
routines that would need to be considered are the routines to order the shoreline and to compute
fetch. The routine to order the shoreline requires that the shoreline be a closed loop. To evaluate
an open stretch of coast in the model, either the landscape domain could be modified to
artificially enclose the open coast or the boundary conditions. The simpler approach is to modify
the landscape domain such that an artificial and large basin was made surrounding the domain,
identifying these as fixed points that do not erode, and making sure the modified landscape is
further than the fetch saturation length from the shoreline of interest. To evaluate an ocean
island, enclose it in land beyond the fetch saturation length in distance from the island. If the
domain is modified such that the shoreline is a closed loop, all routines should function
appropriately. However, if a different routine to order the shoreline is used, the fetch
computation would need to be slightly modified. Currently, fetch is computed as an extended ray
from a shoreline cell that advances at some interval length until it reaches land and allows for a
fetch threshold at some length of wave saturation (See Section 3.3.1). The truncation of
computed fetch at the threshold length is implemented following the calculation of the fetch
length. If there isn't land on the opposite side of the ray, an error would occur. Therefore, by
truncating the fetch length as the ray is extending rather than after the opposite land is found,
fetch could be calculated for open coasts. A more complicated, but preferable approach would be
to change the boundary conditions. If the boundaries of the open stretch of coast were periodic,
the entire coast could retreat without introducing an artificial boundary edge and a larger domain.
The shoreline would be "closed" when it crosses the periodic boundary and arrives at the
repeated point.  If a fetch vector went off the periodic boundary, it would wrap around to the
other side and continues. If a periodic boundary condition is deemed inappropriate, a mirrored
boundary could be used instead. The shape of the coast would be reflected in each boundary, and
fetch vectors would reflect off the boundary.
As a reduced-complexity model, NEWTS1.0 can be applied to investigate coastal
systems in remote environments where field work is difficult or impossible. This includes
locations such as the arctic or Saturn's moon Titan, home to the only other active coastlines in
our solar system. The simplicity of our model allows for efficient, long-term simulations of
coupled landscape evolution and coastal erosion in detachment-limited systems. Among coastal
systems on Earth, investigations of fetch dependence and the resulting morphology given a
combination of erosional mechanisms would be particularly relevant to the carbonate
geomorphology community, as dissolution and wave activity are both often acting
simultaneously along these coasts.

**Acknowledgments**
We thank David Mohrig, Di Jin, Heidi Nepf, Jorge Lorenzo-Trueba, Santiago Benavides,
and Paul Corlies for helpful discussions. Any use of trade, firm, or product names is for
descriptive purposes only and does not imply endorsement by the U.S. Government.
**Funding:**
National Science Foundation Graduate Research Fellowship grant 1745302 (RVP)
NASA Cassini Data Analysis Program grants 80NSSC18K1057 and 80NSSC20K0484
(RVP, JTP, ADA, JMS, SPDB, AGH).
United States Geological Survey, Coastal and Marine Hazards Research Program (RVP)
Heising-Simons Foundation (SPDB)
**Author contributions:**
Conceptualization: RVP, JTP, ADA, JMS, SPDB, AGH
Methodology: RVP, JTP, ADA, JMS
Investigation: RVP, JTP, ADA
Visualization: RVP
Supervision: JTP, ADA, AGH
Writing—original draft: RVP
Writing—review & editing: RVP, JTP, ADA, JMS, SPDB, AGH
**Competing interests:** Authors declare that they have no competing interests.
**Code/Data availability:** NEWTS1.0 model (Palermo et al., 2023) code is available at
https://doi.org/10.5066/P9Q6GDGP.

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
