# Peer review of "NEWTS1.0: Numerical model of coastal Erosion by 1 Waves and Transgressive Scarps 2"

_Geoscientific Model Development, 2023_

## Author Response (AR1)

**Response to Lazarus review:**

I enjoyed reading this contribution by Palermo et al. – the manuscript is clear and carefully conceived, and the model approach and design is engaging.

Inspired by – or riffing on – Howard's (1995) simulation work on escarpment planforms, the authors present an exploratory numerical model of shoreline erosion in rocky coastal landscapes through two mechanisms: uniform retreat, as in Howard (1995); and fetch-limited wave erosion. The example domains showcase the model dynamics for a generic inland lake, but are equally applicable to rocky coastal settings at larger scales. While the model does not explicitly address wave-driven sediment transport, it does represent allogenic controls such as substrate strength, and environmental forcings such as sea-level fluctuations over centuries to thousands of millennia.

> *Thank you for the thoughtful and constructive review!*

My comments here are relatively minor, but I hope may be useful for the authors to consider.

L54 (Section 2.1.1) – Suggest flipping the two paragraphs in this section: lead with Howard, then the others cases. Strikes this reader as more intuitive, and is more aligned with relative conceptual emphasis of the overall work.

> *Suggestion implemented.*

L71 (Section 2.1.2) – Suggest leading with wave-driven erosion, then uniform erosion – which would then mirror the framing in the Introduction (L36).

> *Suggestion implemented.*

L121 (Section 3.1) – mix of "liquid" and "lake" terminology here and throughout manuscript – suggest authors stick with "liquid" as default convention, and then invoke lakes when/if a specific example requires. This more generic terminology might help dispel any confusion about what this model simulates: it's more than a lake model; it could be inland sea, or configured to be an extended reach of open ocean coastline. Similarly, check uses of "sea-level" and "lake-level" fluctuations? ("Liquid level" doesn't work, but a compound phrase like "sea- or lake-level" might serve.)

> *Manuscript text modified for consistency in using "lake" to define a "liquid body" and use of "sea- or lake- level" where appropriate.*

Fig. 3 (and relevant references to "4/8-connected") – It took me a couple of reads and a hard stare at the figure to understand what's meant by this shorthand. If I understand it correctly: "4-connected" determines if a land cell borders a liquid cell on any side; "8-connected" determines if a land cell borders a liquid cell on any side or vertex. A straightforward statement of this mechanics logic at its first mention would be helpful.

> *Definition of 4- and 8- connected added to Sections 3.1.1.2 and 3.1.2. For clarity, we changed the definition of 4-connected to be shoreline cells that are connected to other shoreline cells across*

*an edge and 8-connected to be shoreline cells that are connected to each other across an edge or vertex.*

Fig. 6 (L341) – I found this use of real settings compelling, and I suggest making this the first figure of the paper, because it grounds the whole exercise in a clear physical example. The authors could even expand upon this figure by another two or three panels – a couple more and different examples to demonstrate to the reader that although the manuscript is illustrated using a particular model domain, there's no reason a subset of it could not be set up for an open rocky coastline (e.g., West Coast of USA), another planet (as discussed in the text).

> *Agreed- suggestion implemented. We added a motivational figure to the Introduction, including an open rocky coastline in Scotland and a planetary example on Titan. Fig 6 (now Fig 7) was modified to only show two examples, one wave-eroded and one uniform-eroded coastline.*

L323 – "Because the long-term retreat of bedrock coastlines is generally too slow to be measurable with historical aerial and satellite images, the data needed to fully validate this model are not presently available." I understand this point – but I also wonder whether the authors might speculate on a dimensionless metric (perhaps like a Péclet number, given who the authors are) that reflects the difference in planform shape described in Fig. 5, since this work likewise sets up a conditional "continuum" between uniform and wave-driven end-members. I'd imagine that the examples offered in Fig. 6 would reflect some kind of range in such a metric – and any such metric would be a departure from those offered by Howard (1995). A full exploration would inevitably require a reasonably large volume of shoreline cases, and I'm sure the results would be noisy – but interesting. That undertaking is beyond the scope of this paper, which is necessarily model-forward – but the allusion to future work in that direction is clear. (For what it's worth, I can imagine exemplar settings would need to be chosen judiciously, since fetch-limited, enclosed water bodies dominated by wave-driven sediment transport – e.g. Ashton et al. 2009, cited in the manuscript – can yield very smooth circular lakes that have nothing to do with a uniform erosion condition.)

> *While an exploration of the suggested continuum of shoreline shape is beyond the scope of the present manuscript, we do currently have a paper in review that uses the numerical model to calibrate a quantitative procedure for distinguishing between different dominant shoreline erosion mechanisms. We added more context to the present paper on possible shoreline morphologic analyses in the second to last paragraph of Section 3.4.*

And a final, general note: I would encourage the authors (and/or USGS) to consider the eventual release of this model in Python, for greater accessibility.

> *Thank you, this is an eventual goal for this work.*

**Response to Malatesta review:**

Dear Editors, dear Authors,

Palermo and colleagues introduce a new numerical model for the erosion of lake shorelines in 2D. The model relies on two separate erosion modes that can be selected alternatively or combined: uniform erosion whereby exposure so water level causes shoreline to retreat at a constant rate, and a erosion from waves where wave energy is assumed proportional to the area of fetch faced by the coast. Fetch is calculated as a fraction of the total area in the line of sight of each lake shore point. Continuous erosion of a discrete grid is achieved by attributing a total strength to each pixel that gets progressively weakened until erosion. The shoreline on the square grid can be solved by considering edges, or edges and corners of each pixel. The latter is shown to reduce the total error when compared to an analytical solution. That error estimate from analytical solution is performed elegantly by solving the evolution of a circular lake.

The model is described efficiently and clearly and the accompanying figures support the authors' points rather well. I recommend some moderate revisions to the text to clarify a few key points. I think that I am part of the target audience for this model — I am interested in implementing it — and below I list my main concerns.

Lake versus ocean

It is only in section 3 that I realized you were exclusively targeting lakes (l. 114, 161–162). The discussion of previous models and the importance of coastal erosion (sections 2 and 1) led me to expect a model designed to simulate ocean coastlines, i.e. facing vast open water. The mention of Howard (1995) lake model is the only lacustrine environment present in the review I believe. All other articles focus on open basin settings. It would be beneficial to indicate as early as possible that lake environments are targeted here.

> *We updated the closing paragraph of the introduction to note the emphasis on closed liquid bodies, such as lakes and inland seas, early in the manuscript. We also add to the early explanation that future work could modify the model to apply it to open ocean coasts in the same paragraph.*

The lake versus ocean question matters for the calculation of fetch. I believe that most of the community interested in modeling coastal erosion does so along the ocean. What would be the necessary steps to adapt your model to Hawaii for example. Should fetch simply max out to a given value? For the matter, how would you handle very large closed basins such as the Caspian Sea. Currently it seems that the calculation of fetch is valid for bodies of water that are max a few 10's of km wide. I do not expect the addition of an ocean fetch, or a "very large lake" calculation but it would be good to write a few sentences about the matter. Same would go for the consideration of prevalent winds. What would need to be changed?

> *The model currently includes a saturation length scale for fetch (added a description of this to Section 3.3.1), so that it may be appropriately applied to large bodies of water. We added a paragraph to Section 5 with suggestions of how this model might be modified for open ocean coasts.*

In the review of models, little is said about the timescale they seek to simulate. It would be useful for the reader to have more information about this. This would also situate your own contribution more clearly. You describe it as long term (1–10's of kyr) on line 107. Could it be easily used for a landscape-scale "long term" of 100's of kyr to 1 Myr as well?

> *Yes, with the caveat that erosion rates and sea- or lake- level change would need to be accounted for in the simulation appropriately. We modified this sentence in Section 3 to reflect this.*

When I was reading about the 4- or 8-connected cells, I was wondering how a hexagonal grid would behave. Is it something that you tried? (that comment does not need a modification of the manuscript, this is just my curiosity)

> *We did consider a hexagonal grid, but did not attempt to implement it. Our intuition is that there may be slightly less error in a hexagonal grid, but because the error on the two-dimensional grid was reasonable, we didn't pursue it. As with all structured grids, a hexagonal grid may also yield artifacts in 60 degree orientation which would need to be tested.*

Regarding grid size: In a recent article about the interpretation of marine terraces, we suggested that it is more interesting to track the total duration of sea level occupation at different elevations rather than the unique elevations of high stands (Malatesta et al., Geology, 2022). I look forward to seeing if and how this idea is supported by your model once lake level changes. At the moment, you only present results with constant lake levels. It is likely that most implementation of your model will use varying lake level (or sea level if the model is ported to an ocean coast). As I am thinking about how I would use your work, I wonder about the dimensions of Dx and Dy once the lake level changes. Dx and Dy should be picked such that they satisfyingly capture the coastal geometry. If the coast is relatively smooth, then Dx and Dy can be relatively large. Should I care differently about their dimensions once lake level changes and the coastline jumps at each time step?

> *Yes, the Dx and Dy should be picked such that the coastal features of interest are captured. If there will be lake- or sea-level changes, that should definitely be considered. The new shoreline at a different lake level would be a function of the topography, so the resolution of the grid should be sufficient to capture morphologic variability in the landscape being modeled. Text to this effect was added to the manuscript in Section 3.1.2. Note, however, that the inclusion of strength in the erosion criterion—and the persistence of strength loss between sea-level stands separated in time—is meant to reduce the sensitivity of shoreline retreat to cell size, including in scenarios with sea-level change. This component of the model is consistent with the reviewer's point that the duration of sea level occupation is important.*

In Figure 4 you show how fetch is calculated using a 4-connected scheme. How is it done for the exposed corners of an 8-connected scheme? Are you still using a cosine and go 90º each side of the azimuth, or do you stretch the sampling to 270º.

> *For either the 4-connected case or the 8-connected case, we only consider waves that approach from 90º or less in either direction because, beyond that, diffraction would have to be considered in order to estimate the impact of these waves on the coastline, and, even if so, relative wave energies would be small. For this reason, our approach only considers deep water waves that directly impact the coast. An explanation of this was added to Section 3.3.1 and Section 5.*

In Figure 4 as well: Is Dy longer than Dx for this simulation? It looks like it in the inset b). On line 226 you say that "in practice, we choose Dx to be equal to Dy." If Dx and Dy are not the same as suggested in Figure 4, is the error accordingly stretched in the direction of the longer spatial step? This point is not discussed in section 4.2.1.

*Good catch. The figure was distorted. Dx and Dy are equal in this simulation. Panel b of the figure was corrected to display equal axes.*

*Otherwise, the model would still work with different Dx and Dy, but there could be some differences in error which haven't been tested. This point was added to Section 3.1.2.*

Attributing a strength to each pixel and letting it decay under wave erosion makes sense, and it is a nice solution to the discrete pixels (3.1.3). As I understand, the strength of each cell is independent from its elevation. A cliff is as easy to erode as a low platform. This is standard in most models and not a weakness, but it would be worth pointing it out.

*Suggestion implemented. We included elevation and planform space in the explanation of strength initialization in Section 3.1.3.*

A handful of additional scattered comments:

l. 110 "and" instead of "but"? If the coastline is already straight, the model should not develop cuspate points, correct?

*Correct. Suggestion implemented.*

l. 262–266 Erosion rate is directly proportional to wave energy, but don't you need a wave period, and wave speed, to represent the pace at which that energy is delivered to the coastline? Is that implicitly done when you move to from equation 4 to equation 7?

*The wave period and speed are not accounted for in this equation. We consider the wave energy density as the cross-shore directed wave energy per meter alongshore, which depends on wave height and fluid properties. The cosine term accounts for the "per meter alongshore" part of the energy density. The previous version of the paper inaccurately described this as wave energy flux, which would be E\*cg, whereas we use wave energy, E. By using wave energy, we implicitly assume a single wave period, which means that the rate of energy delivery is included in the rate coefficient. For wave period to be considered, wave energy could be replaced with wave power. The description of wave energy was updated here and a note of the limitation of using wave energy rather than wave power was added to the discussion.*

l. 323–325. A manuscript currently in review at E-surf does this exact measurement using cosmogenic nuclides: https://doi.org/10.5194/egusphere-2023-3020.

*Thank you for pointing us in this direction. A reference to this manuscript and to Hurst et al., 2016 was added in Section 3.4, where we suggest that using such data, localized validation of retreat rates could be done in future work. More detailed testing of the model would require measurements of long-term changes in plan-view shape at a broader scale (or reconstruction of past shape), and such measurements are not currently available.*

Overall, I enjoyed reading this manuscript and found the technical aspects quite clear. The aims of the model (lake versus ocean) should be stated more clearly, and earlier in the text. Many if not most readers will be interested to implement the model facing an open ocean. A brief outline of the necessary modifications would be very advantageous in that regard. Finally, a discussion of the behavior under changing lake level would be welcome as well (role of Dx, Dy).

> *We added context for the closed basins in the abstract and introduction. We also added a paragraph to the discussion describing some possible modifications to implement the model on an open-ocean facing coast. If you end up implementing the model in the future and would like to extend its capabilities beyond those described in the present paper, please reach out. We would be happy to help. Thank you for the thoughtful and constructive review!*

If any of my comments were themselves confusing, please contact me.

Best of luck to the authors,

Luca Malatesta

**Editor comment:**

Based on the constructive comments of both referees, for which I heartily thank them, I encourage Dr. Palermo to not hesitate in preparing a revised manuscript. The authors should be able to submit this revised manuscript draft immediately after closing the "interactive discussion" phase with their response(s) to referee comments.

> *Thank you. Our responses to the reviewer comments have been posted and the revised manuscript is ready for submission pending USGS approval.*

---

## Author Response (AR2)

Dear Dr. Palermo and co-authors,

Following your positive reviews and thorough response to referee comments, I thought it best and most expedient to not return your paper for review. What I have done, therefore, is gone over the changed text myself. Most of these comments are grammatical/etc.

Once you submit the lightly revised manuscript, we should be ready to bring it through towards publication.

Congratulations on your acceptance and thank you for providing this coastal erosion tool to our community.

With best wishes,
Andy
* * *
Line-by-line notes

44-46. Check subject–object agreement. A lake isn't an instance of dissolution/backwasting: it is a lake! A different verb might fix this. Perhaps "Instances of... occur on/at ... "?

> *Changed to: "Instances of uniform erosion on lakes include dissolution and backwasting occurring on karst lakes found in Florida, USA (Fig. 1a) as well as scarp retreat due to weathering and backwasting occurring on Caineville Mesa, Utah, USA (Fig. 1b)."*

70. No hyphen needed for "rocky coastline" because "rocky" is already an adjective.

> *Hyphen removed.*

115. "closed-basin liquid shoreline" makes it seem like the shoreline is liquid. Hm....
I bet that this can be fixed by changing it to a more active-voice reading.
"...subaerial system, the same process law can describe uniform shoreline erosion along the margins of a liquid-filled closed basin"
Something like that?

*Changed to: "Although Howard's model was designed for a different, subaerial system, the same process law can describe uniform shoreline erosion along the margins of a liquid-filled closed basin, as we assume the planform shoreline also erodes at the same rate in all directions."*

184. "lake level in closed liquid bodies" --> "lake level in closed basins"?

*Changed accordingly.*

186. Grammar. To keep it most like what you have:
...bodies. "Lake cell refers to... and lake level..."

*Changed to: "For simplicity, in this manuscript we will use "lake" to refer to the liquid bodies. "Lake cell" refers to cells occupied by liquid and "lake level" refers to the elevation of the liquid level."*

268. Do you mean "differences in error" (from what?) or "error" more generally, or... ?

*Changed to: "resulting errors"*

287. The domain is just the domain, and so it can't have a strength. You model a material with uniform strength across the domain, or?

*Changed to: "We model a material of uniform strength in both planform space and elevation across the domain, but this could easily be extended to a scenario with a material of heterogeneous strength across the domain."*

356. A modeling approach

*Changed to: "The modeling approach presented here does not consider the effects of shoaling or refraction, so waves that would approach the shoreline from beyond 90° are not considered."*

547. "routines" repeated

*Changed to: "The two model algorithms that would need to be considered are the routines to order the shoreline and to compute fetch."*

563-570. I am not quite sure that I understand how this periodic boundary would work, since presumably, the land portion of the domain would repeat as well. Or perhaps I just haven't read carefully enough! Anyway, it's a thought.

> *Changed to: "If the boundaries of the open stretch of coast were periodic in the alongshore direction, the entire coast could retreat without introducing an artificial boundary edge and a larger domain. If a fetch vector went off the periodic boundary, it would wrap around to the other side and continue. If a periodic boundary condition is deemed inappropriate for a specific model task, mirrored boundaries in the alongshore direction could be used instead."*